# Deciphering Genetic Alterations of Hairy Cell Leukemia and Hairy Cell Leukemia-like Disorders in 98 Patients

**DOI:** 10.3390/cancers14081904

**Published:** 2022-04-10

**Authors:** Elsa Maitre, Cécile Tomowiak, Benjamin Lebecque, Fontanet Bijou, Khaled Benabed, Dina Naguib, Pauline Kerneves, Edouard Cornet, Pierre-Julien Viailly, Jeffrey Arsham, Brigitte Sola, Fabrice Jardin, Xavier Troussard

**Affiliations:** 1INSERM U1245, Normandie University, UNIROUEN, UNICAEN, 14032 Caen, France; maitre-e@chu-caen.fr (E.M.); cornet-e@chu-caen.fr (E.C.); pierre-julien.viailly@chb.unicancer.fr (P.-J.V.); brigitte.sola@unicaen.fr (B.S.); fabrice.jardin@chb.unicancer.fr (F.J.); 2Laboratory of Hematology, University Hospital Caen, Avenue de la Côte de Nacre, CEDEX, 14033 Caen, France; naguib-d@chu-caen.fr (D.N.); kerneves-p@chu-caen.fr (P.K.); 3Structure Fédérative SF-MOCAE, University Hospital Caen, Avenue de la Côte de Nacre, CEDEX, 14033 Caen, France; 4Department of Hematology, University Hospital Poitiers, 2 rue Milétrie, 86000 Poitiers, France; cecile.tomowiak@chu-poitiers.fr (C.T.); jeffarsham@yahoo.fr (J.A.); 5Laboratory Hematology, University Hospital Clermont-Ferrand, 58 rue Montalembert, 63000 Clermont-Ferrrand, France; blebecque@chu-clermontferrand.fr; 6Department of Hematology, Institut Bergonié, 229 Cours de l’Argonne, CEDEX, 33079 Bordeaux, France; f.bijou@bordeaux.unicancer.fr; 7Department of Hematology, Centre Hospitalier Public du Cotentin, 46 rue du Val de Saire, 50100 Cherbourg-en-Cotentin, France; benabed-k@chu-caen.fr; 8Department of Hematology, Centre Henri Becquerel, 1 rue d’Amiens, CEDEX, 76038 Rouen, France; 9Hematology Institute, University Hospital Caen, Avenue de la Côte de Nacre, 14033 Caen, France

**Keywords:** hairy cell leukemia, HCL, hairy cell leukemia-like disorders, hairy cell leukemia variant (vHCL), splenic diffuse red pulp lymphoma (SDRPL), genetic alterations, *BRAF*, *MAP2K1*, *KLF2*

## Abstract

**Simple Summary:**

The diagnosis of hairy cell leukemia (cHCL) and HCL-like disorders, including the variant form of HCL (vHCL) and splenic diffuse red pulp lymphoma (SDRPL) can be challenging, particularly in complex situations. The integration of all data, including molecular data, is essential for distinguishing the different entities. The BRAF^V600E^ mutation is identified in most cHCL cases, whereas it is absent in vHCL and SDRPL. *MAP2K1* mutations are observed in half of vHCL cases and in cHCL *BRAF^WT^* and they are associated with a worse prognosis. The interest in deep sequencing for the diagnosis and prognosis of hairy cell leukemia and HCL-like disorders is essential. Some *KLF2* genetic alterations have been localized on the AID consensus motif, suggesting an AID-induced mutation mechanism. *KLF2* is the second most altered gene in HCL, and mutations must be investigated to confirm whether AID could be responsible for the genetic alterations in this gene. Clonal evolution can be observed in half of the cases.

**Abstract:**

Hairy cell leukemia (cHCL) patients have, in most cases, a specific clinical and biological presentation with splenomegaly, anemia, leukopenia, neutropenia, monocytopenia and/or thrombocytopenia, identification of hairy cells that express CD103, CD123, CD25, CD11c and identification of the V600E mutation in the B-Raf proto-oncogene (*BRAF*) in 90% of cases. Monocytopenia is absent in vHCL and SDRPL patients and the abnormal cells do not express CD25 or CD123 and do not present the *BRAF^V600E^* mutation. Ten percent of cHCL patients are BRAF^WT^ and the distinction between cHCL and HCL-like disorders including the variant form of HCL (vHCL) and splenic diffuse red pulp lymphoma (SDRPL) can be challenging. We performed deep sequencing in a large cohort of 84 cHCL and 16 HCL-like disorders to improve insights into the pathogenesis of the diseases. *BRAF* mutations were detected in 76/82 patients of cHCL (93%) and additional mutations were identified in Krüppel-like Factor 2 (*KLF2*) in 19 patients (23%) or *CDKN1B* in 6 patients (7.5%). Some *KLF2* genetic alterations were localized on the cytidine deaminase (AID) consensus motif, suggesting AID-induced mutations. When analyzing sequential samples, a clonal evolution was identified in half of the cHCL patients (6/12 pts). Among the 16 patients with HCL-like disorders, we observed an enrichment of *MAP2K1* mutations in vHCL/SDRPL (3/5 pts) and genes involved in the epigenetic regulation (*KDM6A, EZH2, CREBBP, ARID1A*) (3/5 pts). Furthermore, *MAP2K1* mutations were associated with a bad prognosis and a shorter time to next treatment (TTNT) and progression-free survival (PFS), independently of the HCL classification.

## 1. Introduction

Hairy cell leukemia (HCL) is a rare B-cell chronic lymphoproliferative disorder, with an incidence rate adjusted to the American population of 0.7/100,000 in males and 0.3/100,000 in females [1] and adjusted to the world population of 0.5 and 0.1/100,000 in France [2]. HCL is a well-defined entity characterized by the presence of hairy cells in the peripheral blood and/or bone marrow. The abnormal lymphoid cells are small to medium with homogeneous, spongy, ground-glass chromatin and absent or inconspicuous nucleolus. At diagnosis, the classical form of HCL (cHCL) is characterized by infections, splenomegaly, anemia, neutropenia, monocytopenia or thrombocytopenia. The HCL immunological score, based on the expression of CD103, CD123, CD25 and CD11c (one point for each marker in case of positivity) is high, usually 3 or 4 [3]. The *BRAF^V600E^* mutation in the B-raf proto-oncogene (*BRAF*) gene is detected in up to 80–90% of cHCL cases [4]. The absence of mutation (*BRAF^WT^)* is associated with unmutated (UM) immunoglobulin heavy chain variable region genes (*IGHV*), *VH4-34* rearrangements activating mutations in the mitogen-activated protein kinase kinase 1 (*MAP2K1*) gene and a poor response to nucleoside purine analogs (PNAs) [5].

cHCL has to be distinguished from HCL-like disorders, particularly splenic B-cell lymphoma/leukemia unclassifiable including two provisional entities: the variant form of HCL (vHCL) and the splenic diffuse red pulp lymphoma (SDRPL) [6]. In vHCL, circulating abnormal lymphoid cells are hybrid with intermediate morphology between prolymphocytes and hairy cells. In contrast, a large proportion of small to medium-sized villous lymphoid cells with a polar distribution of the villi and a small or not visible nucleolus are present in the peripheral blood of SDRPL patients. The HCL immunological score is low: 0, 1 or 2. The distinction between the two entities can be challenging due to overlap in their clinical and biological presentations. *MAP2K1* and cyclin D3 (*CCND3*) genetic alterations are observed in 20–40% and 10–20% of cases, respectively [7,8,9,10], the deletion of the long arm of chromosome 7 (del7q) in 20% of cases and a similar *IGHV* repertory with *VH4-34* rearrangements in 20–40% of cases [8,11,12,13]. Finally, the differences are based on cytologic criteria, with a prominent nucleolus and a worse prognosis in vHCL [14]. The relationship between vHCL and SDRPL remains uncertain. The most appropriate terminology for these entities and the precise diagnostic criteria has yet to be established. Based on a limited number of studies, SDRPL could be the first step before the occurrence of vHCL. 

Splenic marginal zone lymphoma (SMZL) with circulating villous lymphocytes is different and develops in the white pulp of the spleen with a biphasic picture. Heterogeneity in blood morphology is common with monocytoid and plamacytoid differentiation. The abnormal lymphoid cells with round nuclei, condensed chromatin and basophilic cytoplasm present polar short villi (so called “villous lymphocytes”) [15,16,17].

Somatic hypermutation (SHM) and class-switch recombination (CSR) are critical physiologic events in an effective normal B-cell immune response. Both SHM and CSR are initiated by activation-induced cytidine deaminase (AID). The presence of AID off-target mutations can also participate in the progression of some B-cell chronic lymphoproliferative disorders such as chronic lymphocytic leukemia [18]. 

The aim of this study was to describe the genomic alterations of a large cohort of 98 patients, 84 with HCL and 16 with HCL-like disorders.

## 2. Materials and Methods

### 2.1. Patients and Samples

We retrospectively investigated 98 patients, 82 with HCL and 16 with HCL-like disorders corresponding to 135 samples (median: 1 sample per patient (min: 1; max: 4)). In order to investigate the clonal distribution of hairy cells between blood and bone marrow, we analyzed blood and bone marrow samples at the same time in 15 patients. We also tested sequential samples in 12 patients to evaluate the evolution of mutations over time. The characteristics of the 98 patients are listed in Table 1 and Appendix A. The cHCL, vHCL and SDRPL diagnoses were based on the 4th edition of the WHO 2017 classification of tumors of the hematopoietic and lymphoid tissues [19]^,^ integrating clinical, morphologic and immunophenotypic criteria. 

All samples of patients with cHCL-BRAFWT, vHCL/SDRPL or HCL-like NOS samples were reviewed in the different diagnostic centers and centrally in Caen. We classified the 98 patients in three subgroups. Group 1 (HCL) included patients with a typical HCL morphology and an immunologic score ≥ 3 (Appendix A), group 2 (vHCL/SDRPL) included patients with either vHCL or SDRPL morphologic criteria or immunologic score ≤ 3 (Appendix A) and group 3 (HCL-like not otherwise specified (NOS)) including patients who did not meet the criteria for groups 1 or 2 (Appendix A). A representative cytomorphology of the different HCL subgroups is available in Appendix A. 

Ethical procedures were observed in accordance with the policy of the CHU de Caen Normandie. 

### 2.2. Next Generation Sequencing (NGS)

The peripheral blood mononuclear cells (PBMC) fraction was collected after gradient density separation (Histopaque^®^, Sigma-Aldrich, Saint-Quentin-Fallavier, France). DNA was extracted with the automated device MagnaPur^®^ (Roche Lifescience, Basel, Switzerland) according to the manufacturer’s recommendations. Library design was performed with the Ion Ampliseq Designer^TM^ software (https://www.ampliseq.com/, accessed on 17 December 2017). The DNA library (Ion Ampliseq^TM^ Library kit), template preparation/chip loading (Ion Chef^TM^ system + Ion PGM^TM^ Hi-Q Chef Kit Reagent) and sequencing (Ion Torrent S5/PGM^TM^, Thermo Fisher Scientific, Waltham, MA, USA) were performed according to the manufacturer’s recommendations. The Trichopanel design and analysis were previously described [20]. The analyzed genes belong to nine functional groups: MAPK signaling pathway (*BRAF*, *MAP2K1*, *DUSP2*, *MAPK15*), epigenetic regulation (*ARID1A*, *ARID1B*, *EZH2*, *KDM6A*, *CREBBP*), cell cycle/apoptosis (*TP53*, *CDKN1B*, *XPO1*), homing (*KLF2*, *CXCR4*), NOTCH pathway (*NOTCH1*, *NOTCH2*), NF-kB pathway (*MYD88*), inflammation (*ANXA1*), splicing (*U2AF1*), differentiation (*BCOR*) and extracellular transport (*ABCA8*), according to published WES data [10,21,22,23]. Data analysis was performed with Torrent suite^TM^ software; then, variant analysis was performed using Torrent Ion reporter^TM^ software. Mutational relevance was analyzed in silico with functional algorithms (SIFT^®^, CADD^®^ and polyphen2^®^), a population database (1000genome^®^, ExAC, GnomAD) and following the recommendations described by Li et al. [24]. Validation of the sequencing quality of the mutations was based on Q-phred score > 20 and a minimum of 100X of deep coverage. 

Determination of % of somatic mutations and repertory of the immunoglobulin heavy variable genes (*IGHV*) was undertaken with the LymphoTrack^®^ Dx IGH FR1 Assay—S5/PGM (Invivoscibe, San Diego, CA, USA) performed according to the manufacturer’s recommendations. 

### 2.3. Statistical Analysis and Data Representation

Data representations were made using R (version 4.4.1: https://www.R-project.org/ accessed on 2 May 2021) and Maftools and GenVisR packages [25,26].

Statistical representations of the Kaplan–Meier test on time to next treatment (TTNT), progression-free survival (PFS) and overall survival (OS) were performed using the Maftools package, and *p*-values were calculated with Log-rank test. TTNT was calculated from the date of diagndosis to ending date of the first treatment to date of second treatment or last patient follow-up. OS was calculated from the date of diagnosis to the date of death or last patient follow-up. PFS was calculated from the date of diagnosis until disease progression, relapse, death or last patient follow-up. Kaplan–Meier curves were compared by log-rank, and median survival of a population was considered 50% probability of survival. *p*-values < 0.05 were considered statistically significant.

## 3. Results

### 3.1. Characteristics of the 98 Patients

Characteristics of the 98 patients are presented in Table 1. The median age of patients at diagnosis was 55 years in cHCL, 72 in vHCL/SDRPL and 74 years in HCL-like NOS. Sixty-four patients were studied at diagnosis and 33 patients at relapse (one patient without unknown status). A mutated (M) *IGHV* profile was observed in 88% of cHCL (30/34 pts), 90% (9/10 pts) of HCL-like NOS and in 40% of vHCL/SDRPL (2/5 pts). The *IGHV* sequences were considered mutated if homology with the closest germline counterpart was less than 98%.

The Trichopanel was relevant for 89 of 98 patients (91%) and detected at least one mutation among the 21 genes studied. In the whole cohort including 135 samples, missense mutations were the most frequent (83%: 183/220); then splicing sit mutations (6%: 14/220) and frameshift deletions (5%: 11/220) (Figure 1A). *BRAF* was the most frequent mutated gene (79%: *n* = 107); then *KLF2* (21% *n* = 28) and *MAP2K1* (8% *n* = 11) (Figure 1B). The median of variants per sample was 1 [min: 1–max: 7] (Appendix A). No single nucleotide variant (SNV) was found in 9 patients for any of the 21 targeted genes (3 cHCL, 5 HCL-like NOS, and 1 patient with vHCL/SDRPL) (Appendix A).

### 3.2. cHCL Has a Specific Molecular Signature 

The *BRAF* mutation was identified in 93% (76/82 pts) of cHCL patients (Figure 1C, Appendix A): most of them were *BRAF^V600E^* but one patient (UPN-50) presented *BRAF^F595L^*. In decreasing order of frequency, the genetic alterations were *KLF2* in 23% (*n* = 19) and *CDKN1B* in 7.5% (*n* = 6). *MAP2K1* mutations were identified in two cHCL patients; both were *BRAF^WT^* with an HCL score ≥ 3. Epigenetic genes (*ARID1A, CREBBP, KDM6A or EZH2*) were mutated in 8 cHCL patients (10%), 3 vHCL/SDRPL (60%) and 1 patient with HCL-like NOS (9%). When analyzing, at the same time, blood and bone marrow in 15 patients, no difference was observed between the two compartments except for UPN-111. BM-derived hairy cells were *BRAF^V600E^*, *KLF2^S275=^* and *CXCR4^C218=^*; whereas PB-derived cells were *BRAF^V600E^* and *TP53^D393G^*. We confirmed through analysis of polymorphisms and *IGHV* repertory that PB and BM samples originated from the same patient. 

### 3.3. Sub-Clonal Mutations and Variant Allele Frequency Changes Were Found at Relapse

The mutational landscape was different if analysis was performed at diagnosis or at relapse. We sequentially analyzed 12 cHCL patients with a median time of 91 months [20.9–173.5] (Appendix A). Half of the patients had changes: 5 patients showed new mutations and 1 patient a loss. *CREBBP* was mutated in 4 samples at relapse vs. 1 at diagnosis with no statistically significant mutational enrichment. No statistically significant difference was demonstrated for the entire cHCL cohort (Appendix A). Patient UPN-45 is a representative case and was followed for 314 months. He relapsed 6 times and received interferon treatment during five years; then cladribine at relapse 1 and 2 (R1, R2), pentostatin at R3, four cycles of rituximab (R4); vemurafenib, a BRAF inhibitor (R5) and moxetumomab-pasudotox, an anti-CD22 immunotoxin (R6). During the multiple relapses, we observed the emergence of two sub-clonal *KLF2* mutations that disappeared at R3 and an amplification of the clone containing *CREBBP* and *EZH2* mutations (Figure 2).

### 3.4. KLF2 Is a Gene of Interest 

*KLF2* is the most altered gene in cHCL after the *BRAF^V600E^* mutation. Indeed, 19 patients presented missense of splicing mutations. Missense mutations were localized in two specific domains: the zinc finger domain (amino acid 267 to 350) and the putative nuclear localization signal (amino acid 254 to 274) (Figure 3A). In patients who carried *KLF2* missense mutations, we also found multiple intronic or synonymous mutations increasing the number of patients with *KLF2* mutations from 19 to 30 (Appendix A). In total, 14 of the 30 patients (19/45 samples) with *KLF2* mutations had other multiple mutations with a mean of 2.0 mutations per sample (including intronic of synonymous mutations). Polymorphisms were excluded with VAF corresponding to tumor infiltration or sub-clonal mutation. Most *KLF2* mutations were G > A (49%: 44/90) or C > T (37%: 33/90) transitions; transversions (C > A, C > G, T > A, T > G) were rare (Appendix A). The multiple mutations and the finding of transitions suggest an activation-induced cytidine deaminase (AID)-mediated process. We therefore investigated whether the AID consensus motif RCY (R = A/G, Y = C/T) was present in *KLF2* mutations. Among the 44 *KLF2* mutations, 9 were localized within the AID consensus motif, particularly mutations involving hotspot amino acid serine at location 275 on the zinc-finger domain (Figure 3B).

### 3.5. Genomic Signature in HCL-like Patients

Among the 16 HCL-like disorders: 5 patients were classified vHCL/SDRPL and 11 patients HCL-like NOS. When comparing with cHCL, vHCL/SDRPL patients had a median of 2 variants per patient, suggesting higher molecular complexity (Appendix A). Genomic features of these patients were different from those of the cHCL cohort: we observed an enrichment of *MAP2K1* mutations in vHCL/SDRPL (3/5 pts) vs. 2/93 including 82 cHCL and 11 HCL-like NOS (*p*-value = 6.368053 × 10^−4^). *KDM6A,* encoding a lysin-demethylase, was also more mutated in vHCL/SDRPL (2/5) vs. 1/93 of the rest (*p*-value = 6.180307 × 10^−3^) (Figure 4). *KDM6A* mutations were deleterious, either frameshift or involving the splice site. Epigenetic genes (*KDM6A, EZH2, CREBBP, ARID1A*) were altered in 3/5 of vHCL/SDRPL. For the one remaining patient (UPN-v6), we found a *U2AF1* hotspot p.S34F mutation and multiple *TP53* mutations during the progression of the disease (UPN-v6E1/UPN-v6E2) (Appendix A). *TP53* mutations were also found in UPN-v13. HCL-like NOS patients had a reduced number of mutations (Appendix A) involving *KLF2* (*n* = 1), *NOTCH1* (*n* = 1), *U2AF1* (*n* = 1), *ARID1A* (*n* = 1), *CDKN1B* (*n* = 1) and *TP53* (*n* = 1) (Appendix A).

### 3.6. Clinical Impact and Prognosis

Only mutations of the *MAP2K1* gene were associated with a poor prognosis for TTNT and PFS (median of TTNT 11.15 months vs. 49.74 months, median of PFS: 9.97 months vs. 46.89 months) independent of HCL classification (Figure 5). A comparison between disease sub-groups (cHCL, cHCL *BRAF^WT^*, vHCL/SDRPL and HCL-like NOS) showed a slight difference between HCL and HCL-like disorders in PFS but not in TTNT (median of PFS: cHCL 48.41 months, cHCL *BRAF^WT^* 46.89 months, vHCL/SDRPL 11.06 months and HCL-like NOS 5.67 months; median of TTNT: cHCL 63.1 months, cHCL *BRAF^WT^* 47.9 months, vHCL/SDRPL 25.8 months and HCL-like NOS 49.7 months) (Appendix A).

## 4. Discussion

We updated and extended the analysis of the mutational landscape of a large cohort of 98 patients with either cHCL or HCL-like disorders, but due of the rarity of the disease only 16 patients made up the HCL-like disorder groups (5 patients with vHCL or SDRPL, and 11 patients with HCL-like NOS disease). cHCL diagnosis was improved through detection of the hotspot *BRAF^V600E^* mutation [21]. However, the frequency of the *BRAF^V600E^* mutation differed according to the studies, ranging from 80% to 100% [4,21]. Some alternative *BRAF* mutations were also reported in exon 11 [27] as well as a *IGH/BRAF* fusion leading to BRAF upregulation [28]. Consistent with the literature, we found the *BRAF^V600E^* mutation in 93% (76/82 pts). Among the 6 *BRAF^WT^* patients, 1 patient (UPN-50) presented an alternative *BRAF* mutation (*BRAF^F595L^* located in exon 15), 2 had *MAP2K1* mutations (UPN-40, UPN-46). In the 3 other cases, no alternative *BRAF* including exons 11 or 15 or *MAP2K1* mutations were detected. All 6 *BRAF^WT^* HCL patients had an immunological HCL score (CD103+, CD123+, CD25+, CD11c+) of 4 in 5/6 cases and 3 in 1 patient (CD123-). The *IGHV* profile was mutated in 3 cases (UPN-10, UPN-75, UPN-91) and unmutated in the 3 others (UPN-40, UPN-46, UPN-55). For the 3 remaining patients (UPN-40, UPN-46, UPN-55), the immunoglobulin repertory was *VH4-34* [4,29]. 

In accordance with the hypothesis that *BRAF* alteration is necessary but not sufficient to induce HC, one third (27/82 pts) of the cHCL patients had more than 1 altered gene among the 21 evaluated. Indeed, the *BRAF^V600E^* mutation was found in the hematopoietic stem cells of HCL patients with VAF increasing in the pre-B compartment [30]. 

As in chronic lymphocytic leukemia [31], the analysis of samples at diagnosis/relapse highlighted the heterogeneity of hairy cells and their sub-clonal evolution during the course of the disease. Although the differences were not statistically significant, the sub-clonal evolution of HCL necessitates realization of a novel analysis of the mutational landscape for better understanding of the disease. Furthermore, the characterization of sub-clonal mutations could be necessary for management of future treatments and to avoid promotion of a sub-clone during the relapse.

*KLF2* is the second most altered gene in cHCL patients and mutations were reported in 10–16% of cases [32,33,34]. In the studied cohort, *KLF2* was mutated in 19 patients or 30 patients (i.e., 45 samples) if we added the follow-up samples with silent and intronic mutations. *KLF2* encoding for Krüppel-like factor 2 is involved in B-cell homing to lymph nodes and inhibition of the NF-kB pathway. *KLF2^275N^*, *KLF2^275T^* and *KFL2^T271I^* mutations were localized within the zinc finger domain or the nuclear localization signal (NLS) and were described in splenic marginal zone lymphoma (SMZL) [32,33]. Piva and coworkers showed that mutations involving the NLS lead to cytoplasmic relocalization of *KFL2* and affect its transcription factor activity [33]. *KLF2* knock-out led to the deregulation of B-cell differentiation and trafficking, but was not sufficient to induce lymphoma in a murine model [35,36]. Thereby, mutations in *KLF2* in HCL patients could explain both the extra-nodal localization of hairy cells and NF-kB pathway upregulation [37]. As described previously, multiple mutations in *KLF2* are frequent [33]. In this cohort, 14/30 patients (19/45 samples) presented multiple *KLF2* mutations. This could be explained by the dependency of *KLF2* mutation on specific oncogenic signatures such as aging, or activation-induced cytidine deaminase motifs. Interestingly, we noted that the pattern of mutations in *KLF2* was suggestive of AID activity. AID encoding by *AICDA* is crucial for immune response. It allowed B-cell receptor maturation and antibody diversification through somatic hypermutation (SHM) and class switch recombination (CSR) [38]. AID induced SHM and CSR by deaminating cytosine residues in the RCY motif (R = A/G, Y = C/T) of immunoglobulin genes, or off-target genes [39]. The generated mismatch led to transition mutations (C > T or G > A, depending on the DNA strand). Furthermore, *KLF2* seems to be preferentially mutated in cHCL rather than HCL-like disorders and could be related to a higher level of expression of *AICDA* [40]. *KLF2* mutations in HCL must be investigated to confirm whether AID could be involved.

In our study, only one cHCL patient (UPN-11) had a *TP53* mutation (*TP53^D393G^*, VAF = 2.2%) and it was detected in the PB-derived cells. The mutation is considered as functional in the IARC database (https://tp53.isb-cgc.org, accessed on 26 February 2022). In turn, it probably has no functional consequences. These data are consistent with previously published data in which the frequency of *TP53* mutations differs considerably from one study to another, ranging from 0–2% [5,41] to 27% [42]. Three *TP53* mutations were found in HCL-like patients (two at diagnosis and one during the progression of the disease). In contrast, *TP53* mutations have been reported in nearly 30% of vHCL cases [5,43]. We reported *TP53* mutations in three HCL-like patients in our cohort; two vHCL/SDRPL (UPN-v13, sample at diagnosis, UPN-v6, sample at evolution) and one HCL-like NOS (UPN-v8, sample at diagnosis). According to the IARC *TP53* database, all mutations found in HCL-like patients were non-functional. 

*CDKN1B* that encodes p27, an inhibitor of cell cycle, was mutated in 11–16% of HCL patients [22,41] and 7% in our cohort. 

We also observed mutations of *MAP2K1* in five patients *BRAF^WT^* (two cHCL, three vHCL/SDRPL). This finding is consistent with previously published data in which 48% of *BRAF^WT^* HCL patients (classical and variant forms) presented *MAP2K1* mutations [10]. The determination of *MAP2K1* status is essential in the management of HCL patients. Indeed, some mutations allowed the use of MEK inhibitors, and others did not. *MAP2K1^K57N^* improves the sensitivity of cells to MEK inhibitors, whereas *MAP2K1^Q56P^*, *MAP2K1^I103N^* and *MAP2K1^C121S^* generate resistance by impairing the allosteric binding of the drug [44,45,46,47]. In our study, mutations of the *MAP2K1* gene were associated with a bad prognosis and a shorter TTNT and PFS, independently of the HCL classification. However, further analyses are needed to confirm in a large cohort of patients the clinical impact of the *MAKP2K1* mutations and to demonstrate if they are associated with confounding factors.

As described previously [20], the alteration of genes involved in the epigenetic regulation, especially *KDM6A,* were recurrent in HCL and HCL-like disorders. We added to the initial cohort one patient with *KDM6A* disruption (frameshift deletion in patient UPN-80). *KDM6A* (also known as ubiquitously transcribed tetratricopeptide repeat protein X-linked or UTX) encodes a lysine demethylase that removes di- and tri-methyl groups from lysine 27 of histone 3 (H3K27). Disruptive *KDM6A* mutations were found in lymphoid neoplasms such as multiple myeloma and T-cell acute lymphoblastic leukemia [48]. Mutations of *KDM6A* result in the loss of the highly conserved C-terminal region (including Jumomji and zinc binding domains) essential for the demethylase activity [49]. Loss of KDM6A activity may sensitize tumor cells to demethylating agents such as EZH2 inhibitors [50]. 

*CREBBP* encoding a histone lysine-acetylase was involved in H3-K27 acetylation whose action was associated with *KDM6A*. *CREBBP* mutations were described in 5.7% (3/53) of cHCL patients and 12.5% (1/8) of vHCL patients [41]. Three cHCL patients (3.7%, 3/82) and one vHCL patient (1/5) had *CREBBP* mutations. 

The *U2AF1^S34F^* hotspot mutation was present in two HCL-like patients, one HCL-like NOS and one vHCL/SDRPL (UPN-v15, UPN-v6) and no cHCL [8,10,41]. These combined data suggest that while HCL-like NOS and vHCL/SDRPL have a similar mutational signature, vHCL could have more genomic instability (median of variants per sample 2 in vHCL/SDRPL and 0 in HCL-like NOS) 

## 5. Conclusions

We confirmed the value of using deep sequencing for the diagnosis and the prognosis of hairy cell leukemia and HCL-like disorders. Looking for genetic alterations is crucial in the field of personalized medicine to improve the best care management. Further studies with a higher number of patients and addition of pertinent altered genes such as *CCND3* or *KMT2C* are required to confirm and complete these results. Especially, *MAP2K1* mutations lead to poorer prognosis in hairy cell leukemia patients and must be confirmed in a cohort with a greater number of patients. 

## Figures and Tables

**Figure 1 cancers-14-01904-f001:**
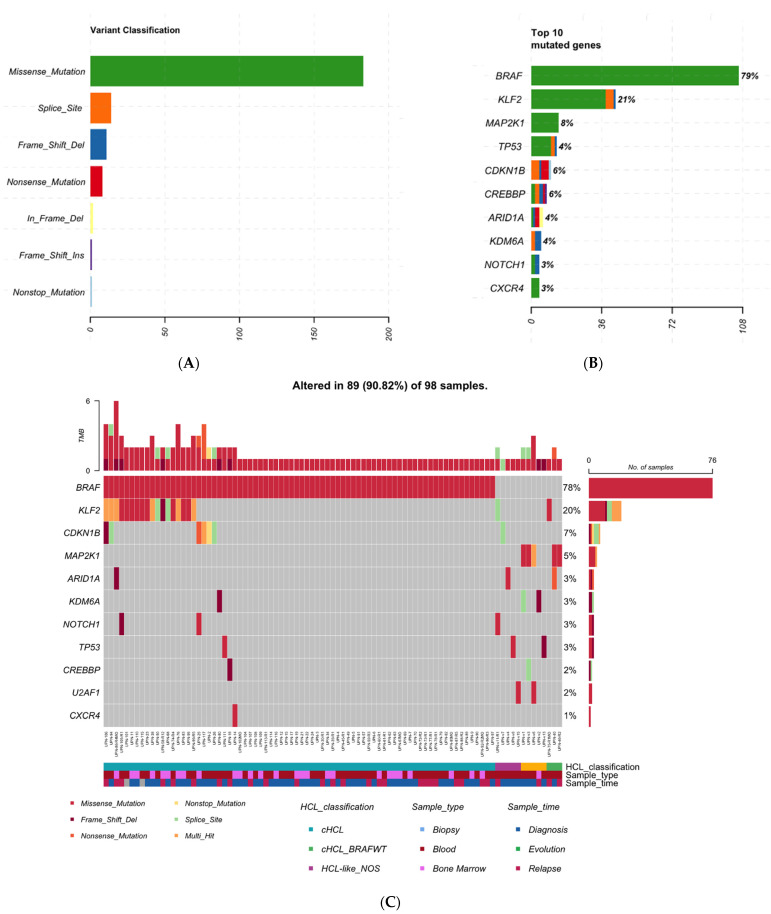
(**A**) Classification of the variants (*n* = 220) in the overall cohort (135 samples). (**B**) Top10 mutated genes in the overall cohort (*n* = 135 samples). (**C**) Waterfall representation for the unique patients (*n* = 98). When several samples were available for one patient, the first sample was used.

**Figure 2 cancers-14-01904-f002:**
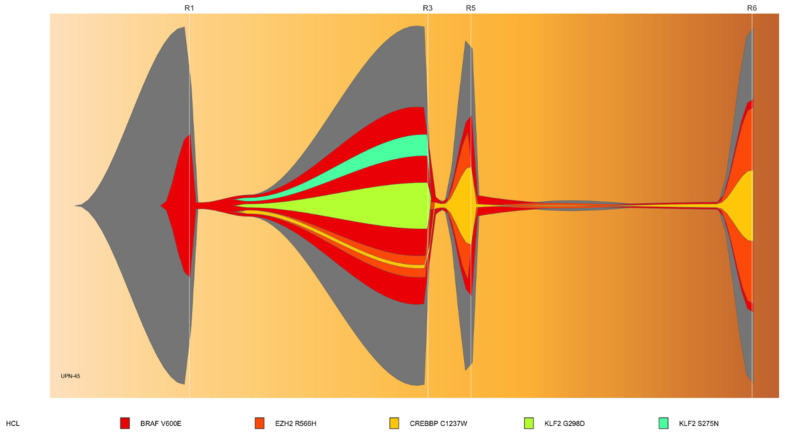
Fish-plot representation of UPN-45 showing clonal evolution of the mutations along the progression of the disease. The patient received interferon during five years; then 2-CDA at relapse 1 (R1) and 2 (R2), pentostatin at R3, 4 cycles of rituximab (R4), BRAF-inhibitor: vemurafenib (R5), and anti-CD22 immunotoxin moxetumomab-pasudotox (R6). No sample was obtained at R2 and R4. R1: Relapse 1 (165 months); R3: Relapse 3 (226 months); R5: Relapse 3 +237 months; R6: Relapse 6 +309 months.

**Figure 3 cancers-14-01904-f003:**
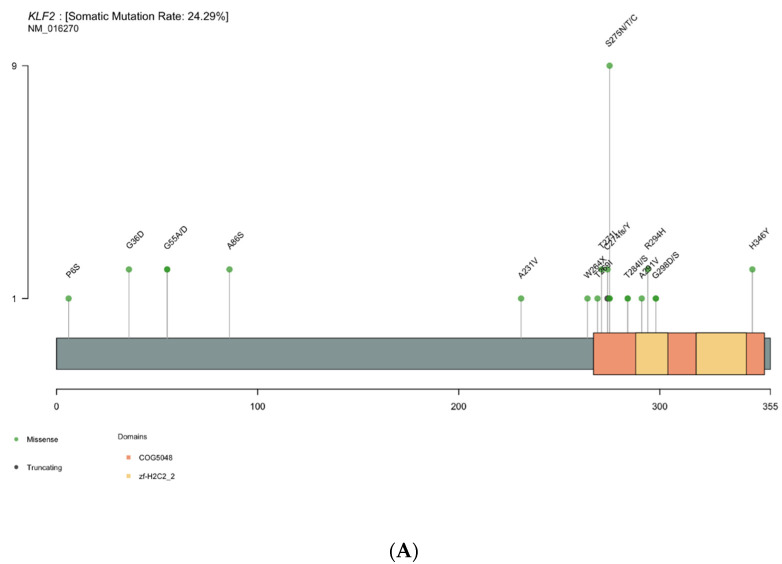
(**A**) Lolliplot representation of *KLF2* mutations, excluding synonymous and deep intronic mutations. In green, missense mutations of the two domains: COG5048, conserved protein domain family COG5048, including zinc-finger proteins and Zf-H2C2_2, zinc-finger double domains. (**B**) transition–transversion plot of the *KLF2* mutation. Intronic and synonymous mutations are included.

**Figure 4 cancers-14-01904-f004:**
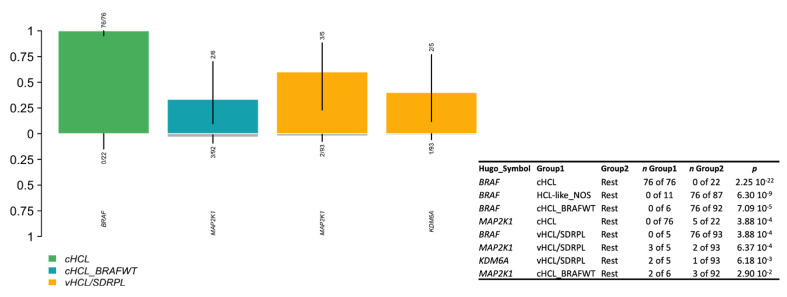
Enrichment of a specific mutated gene in the three different sub-groups.

**Figure 5 cancers-14-01904-f005:**
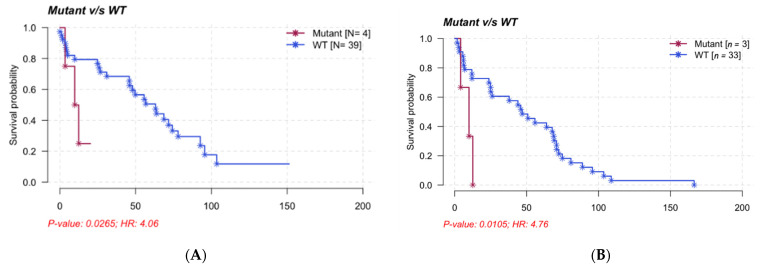
Survival analysis of the *MAP2K1* mutation on TTNT (**A**), PFS (**B**) and OS (**C**), Kaplan–Meier representation, comparison by log-rank.

**Table 1 cancers-14-01904-t001:** Characteristics of 98 patients.

Variate	HCL (*n* = 82)	HCL-like Disorders (*n* = 16)	
		vHCL/SDRPL (*n* = 5)	HCL-like NOS (*n* = 11)
CLINICAL FEATURES			
Sex ratio (M/F)	2.9	5	2.7
Median age (years [min-max])	55.8 [30–82]	72 [58–82]	74 [67–95]
Lymphocytosis (>4 G/L)	12/61	3/4	5/10
Monocytopenia (<0.2)	41/56	0/4	2/10
Diagnosis (*n*, %), relapse (*n*, %) sample	51 (63%), 30 (27%)	4 (80%), 1 (20%)	9 (82%), 2 (18%)
Median number of treatment lines[min-max]	2 [1–12]	1 [0–1]	0.5 [0–1]
BM (*n*, %), PB (*n*, %)	28 (34%), 54 (66%)	1(20%), 4 (80%)	1 (9%), 10 (91%)
IMMUNOPHENOTYPE			
CD11c	72/72	5/5	10/11
CD25 (*n*)	72/72	1/5	2/11
CD103 (*n*)	71/72	4/5	4/11
CD123 (*n*)	65/68	1/4	2/11
GENETIC ALTERATIONS			
*IGHV* mutated (*n*, %)	30/34 (88%)	2/5 (40%)	9/10 (90%)
*IGHV* repertory VH4-34 (*n*)	3/34	3/5	1/10
*IGHV* repertory VH3 (*n*)	15/34	0/5	5/10
*IGHV* repertory VH1-2 (*n*)	2/34	0	2/10

Abbreviations: BM: bone marrow; PB: peripheral blood. Monocytopenia is defined by absolute monocyte count < 0.2 G/L and lymphocytosis by absolute lymphocyte count > 4.0 G/L.

## Data Availability

Data available on request due to restrictions e.g., privacy or ethical. The data presented in this study are available on request from the corresponding author.

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
