# Peer review of "Deciphering Genetic Alterations of Hairy Cell Leukemia and Hairy Cell Leukemia-like Disorders in 98 Patients"

_cancers, 2022, doi:10.3390/cancers14081904_

Round 1
Reviewer 1 Report
-Needs improvements in the English language
-Data needs to be much more clearly presented
-Correlation of mutational panel with clinical outcomes needs to be better emphasized: Most of the paper is enumeration of different findings and observations. Very little space is dedicated to correlating different mutations to clinical outcomes be it survival/TTNT/Relapse etc.
-There is no mentioning of limitations to the study (aside from needing a bigger sample size (mentioned in conclusion) and lacking statistical significance in the data and correlations reported
Author Response
Reviewer 1
-Needs improvements in the English language
English language was revised by a native English person.
-Data needs to be much more clearly presented
Thanks to the remark. As you can see, we improved the text readability in the different sections of the manuscript.
-Correlation of mutational panel with clinical outcomes needs to be better emphasized: Most of the paper is enumeration of different findings and observations. Very little space is dedicated to correlating different mutations to clinical outcomes be it survival/TTNT/Relapse etc.
Thank you for the remark. We demonstrated for the first time that MAP2K1 mutations were associated with a bad prognosis for the TTNT and also PFS. For emphasizing this interesting data, we added in the discussion section:
Lines 346-349: In our study, mutations of the MAP2K1 gene were associated with a bad prognosis and a shorter TTNT and PFS independently of the HCL classification. However, further analyses are needed to confirm in a large cohort of patients the clinical impact of the MAKP2K1 mutations and to demonstrate if they are associated with confounding factors.
-There is no mentioning of limitations to the study (aside from needing a bigger sample size (mentioned in conclusion) and lacking statistical significance in the data and correlations reported
Thank you for the comment and we agree with that. To complete the article, we added in the conclusion some limitations as follows:
Lines 372-377: sequencing for the diagnosis and the prognosis of hairy cell leukemia and HCL-like disorders. Looking for genetic alterations is crucial in the field of personalized medicine to improve the best care management. Further studies with a higher number of patients and addition of pertinent altered genes like CCND3 or KMT2C are required to confirm and complete these results. Specially MAP2K1 mutations lead to poorer prognosis in hairy cell leukemia patients and have to be confirmed in a cohort with a higher number of patients.
Reviewer 2 Report
The reported work is deemed to be of considerable interest but some aspects of the manuscript need to be improved.
The title is appropriate for the content.
Abstract:
The authors use the term “association” to describe the relation between BRAF mutations and KLF2 mutations, but genetic association should be statistically demonstrated and
this result is never reported in the manuscript.
In table 1 check if the number of patients with IGHv mutated is 2 or 3 as reported in VH434 line
Material and Methods:
The authors should explain the decision to analyse more samples for some patients.
Briefly describe the Trichopanel design and analysis
Result:
“BRAFV600E mutation was present in 93%..” should be changed in “BRAF mutation was..”
Clinical impact and prognosis
I feel that the number of patients analysed is not robust enough to confirm the finding of MP2K1 mutation being associated with poor prognosis. In fact, patients MAP2K mutated are only 4; moreover the authors should evaluate if age could be a confounder effect.
Discussion
The discussion should be substantially revised. In this section the authors should discuss the impact of their findings and not report results; results should only be reported in Result section
Author Response
Reviewer 2
The reported work is deemed to be of considerable interest but some aspects of the manuscript need to be improved.
The title is appropriate for the content.
Abstract:
The authors use the term “association” to describe the relation between BRAF mutations and KLF2 mutations, but genetic association should be statistically demonstrated and this result is never reported in the manuscript.
Thank you for this remark. We didn’t make a statistical analysis to demonstrate an association between BRAF and KLF2. We changed the sentence as follows:
Lines 41-42: BRAF mutations were detected in 76/82 patients of cHCL (93%) and additional mutations were identified in Krüppel-like Factor 2 (KLF2) in 19 patients (23%) or CDKN1B in 6 patients (7,5%).
In table 1 check if the number of patients with IGHv mutated is 2 or 3 as reported in VH434 line
Thank you for the comment: after checking, three cHCL patients were unmutated and VH4-34.
A total of 4 cHCL patients was unmutated (three VH4-34 and one not using VH4-34 more, see details in supplementary table 1)
Material and Methods:
The authors should explain the decision to analyse more samples for some patients.
We analyzed both blood and bone marrow samples for some patients. We choosed to analyze both tissues. So far, in our cohort, only one patient presented differences between blood and bone marrow (UPN-111, described lines 194-198)
Furthermore, we also analyzed sequential samples and we previously described some variations in the mutational landscape in 6 of the 12 samples.
We completed in the material section as follows:
Lines 101-105: In order to investigate the clonal distribution of hairy cells between blood and bone marrow, we analyzed blood and bone marrow samples at the same time in 15 patients. We also tested sequential samples in 12 patients to evaluate the evolution of mutations over time.
Briefly describe the Trichopanel design and analysis
As recommended, we added the description in the method section.
Lines 140-143: Mutational relevance was analyzed in silico with functional algorithms (SIFT®, CADD® and polyphen2®), population database (1000genome®, ExAC, GnomAD) and followed the recommendations described by Li et al. [22] . Validation of the sequencing quality of the mutations was based on Q-phred score >20 and a minimum of 100X of deep coverage.
Result:
“BRAFV600E mutation was present in 93%..” should be changed in “BRAF mutation was..”
Thank you for that. We corrected, as follows: Line 188: BRAF mutation was present in 93% (76/82 pts) of cHCL patients.
Clinical impact and prognosis
I feel that the number of patients analysed is not robust enough to confirm the finding of MP2K1 mutation being associated with poor prognosis. In fact, patients MAP2K mutated are only 4; moreover the authors should evaluate if age could be a confounder effect.
Thank you for this comment. We discussed this limitation as follows: Lines 352-355: further analyses are needed to confirm in a cohort with a higher number of patients the bad prognosis of MAKP2K1 mutations and to demonstrate whether they are associated with confounding factors.
Discussion
The discussion should be substantially revised. In this section the authors should discuss the impact of their findings and not report results; results should only be reported in Result section
We modified the discussion as follows: KLF2 mutations in HCL must be investigated to confirm whether AID could be involved. We also changed the place.
And the conclusion was modified as follows:Lines 377-383: We confirmed the value of using deep sequencing for the diagnosis and the prognosis of hairy cell leukemia and HCL-like disorders. . Looking for genetic alterations is crucial in the field of personalized medicine to improve the best care management. Further studies with a higher number of patients and addition of pertinent altered genes like CCND3 or KMT2C are required to confirm and complete these results. Specially MAP2K1 mutations lead to poorer prognosis in hairy cell leukemia patients and have to be confirmed in a cohort with a higher number of patients.
Reviewer 3 Report
This manuscript is a review of similarities and differences between classical hairy cell leukemia, variant hairy cell leukemia, and splenic diffuse red pulp lymphoma. Most interestingly it goes on to provide mutational differences amongst these disorders that can assist in diagnosis.
- Throughout the manuscript the authors mention cHCL. I would initially state that this represents classical Hairy Cell Leukemia to differentiate it from variant Hairy Cell Leukemia. Please keep this consistent as well as vHCL- it sometimes is referred to as HCLv. This is especially important in the Abstract and Summary.
- The first reference to AID should explain that it is the enzyme activation induced cytidine deaminase. As it stands it does not happen until page 7.
- In the methods section, were the bone marrow and peripheral blood samples reviewed centrally or were they reviewed at their respective diagnostic centers?
- Please review the manuscript as there are numerous typos- periods and spaces where they should not be present.
- If there were 98 patients- 82 patients with cHCL and 16 patients with HCL like disorders- why were there 135 samples? Did some patients have more than 1 sample? If samples were collected at relapse it should be clearly described as such.
- There are discrepancies throughout the manuscript in the numbers. It should always read 82 cHCL patients and 16 HCL-like disorders divided into 5 patients with vHCL/SDRPL and 11 patients with HCL-like NOS as depicted in Table 1. Figure S1 has these numbers reversed.
- Why were vHCL and SDRPL samples combined? Was it simply because of numbers? If so then it may be useful to analyze the data based on what is HCL, ie cHCL, versus all other cases.
- KLF2 mutations have been described in HCL- this should be mentioned and appropriately cited (Tiacci et al, JCO 2017 PMID 28297625)
- The HCL score is mentioned in the manuscript but never described. Please add a brief description in the methods since it helped differentiate cases.
Author Response
Reviewer 3
- Throughout the manuscript the authors mention cHCL. I would initially state that this represents classical Hairy Cell Leukemia to differentiate it from variant Hairy Cell Leukemia. Please keep this consistent as well as vHCL- it sometimes is referred to as HCLv. This is especially important in the Abstract and Summary.
Thank you. We changed and checked: HVLv was changed by vHCL in the entire manuscript.
- The first reference to AID should explain that it is the enzyme activation induced cytidine deaminase. As it stands it does not happen until page 7.
We added in the text: In the abstract line 43: Some KLF2 genetic alterations were localized on the cytidine deaminase (AID) consensus and In the introduction lines 91-95: Somatic hypermutation (SHM) and class-switch recombination (CSR) are critical physiologic events in an effective normal B-cell immune response. Both SHM and CSR are initiated by activation-induced cytidine deaminase (AID).The presence of AID off-target mutations can also participate to the progression of some B –cell chronic lymphoproliferative diorders such as chronic lymphocytic leukemia [18].
- In the methods section, were the bone marrow and peripheral blood samples reviewed centrally or were they reviewed at their respective diagnostic centers?
Thank you for asking us this clarification. Indeed, bone marrow and peripheral blood samples of cHCL were reviewed in respective diagnosis center and also centrally in Caen. All patients with cHCL-BRAFWT, vHCL/DSRPL or HCL-like NOS samples were reviewed in respective diagnosis center and centrally in Caen.
We added the clarification as follows:
lines-110-111: All samples of patients with cHCL-BRAFWT, vHCL/SDRPL or HCL-like NOS samples were reviewed in the different diagnostic centers and also centrally in Caen.
- Please review the manuscript as there are numerous typos- periods and spaces where they should not be present.
Thank you. The correction was done
- If there were 98 patients- 82 patients with cHCL and 16 patients with HCL like disorders- why were there 135 samples? Did some patients have more than 1 sample? If samples were collected at relapse it should be clearly described as such.
We analyzed both blood and bone marrow sample for some patients. We choose to analyze both tissues to find if the circulating pool of hairy cells in the peripheral blood is identical to the medullary compartment. So far, in our cohort, only one patient presented differences between blood and bone marrow sample (UPN-111).
lines 192-194. Furthermore, for some patient we also have sequential samples, and we are able to report differences in 6 of the 12 samples.
To follows your advice and improve the readability, we completed in the material section as follows:
Lines 101-105In order to investigate the clonal distribution of hairy cells between blood and bone marrow, we analyzed blood and bone marrow samples at the same time in 15 patients. We also tested sequential samples in 12 patients to evaluate the evolution of mutations over time.
- There are discrepancies throughout the manuscript in the numbers. It should always read 82 cHCL patients and 16 HCL-like disorders divided into 5 patients with vHCL/SDRPL and 11 patients with HCL-like NOS as depicted in Table 1. Figure S1 has these numbers reversed.
Thank you very much. We corrected this mistake as follows:
lines 387-389: Figure S2. A: representation of number of variants per sample of cHCL (n = 82), vHCL/SDRPL(n = 5) and HCL-like NOS (n = 11) individual patients. B: top 10 mutated gene of cHCL (n = 82), vHCL/SDRPL(n = 5) and HCL-like NOS (n = 11) individual patients.
- Why were vHCL and SDRPL samples combined? Was it simply because of numbers? If so then it may be useful to analyze the data based on what is HCL, ie cHCL, versus all other cases.
We combined vHCL and SDRPL because main differences are based on cytologic criteria, with a prominent nucleolus and a worse prognosis in vHCL. The relationship between vHCL and SDRPL remains uncertain. The most appropriate terminology for these entities and the precise diagnostic criteria has yet to be established. Based on a limited number of studies, SDRPL could be the first step before the occurrence of vHCL.
The clarification was presented lines 80-85 in the introduction.
- KLF2 mutations have been described in HCL- this should be mentioned and appropriately cited (Tiacci et al, JCO 2017 PMID 28297625) (Reference 34).
Thank you, we added this important reference.
line 301 : KLF2 is the second most altered gene in cHCL patients and mutations were reported in 10-16% of cases [32–34].
- The HCL score is mentioned in the manuscript but never described. Please add a brief description in the methods since it helped differentiate cases.
The immunological score was introduced in the introduction for improving the readability.
We also completed the sentence as follows: Lines 62-63: The HCL immunological score, based on the expression of CD103, CD123, CD25 and CD11c (one point for each if positive) is high, usually 3 or 4 [3].
Round 2
Reviewer 3 Report
The authors have done a nice job of explaining the mutational landscape of hairy cell leukemia and its variants. The have also systemically reviewed all of my comments with adequate responses.
Major discussion
1. In the discussion the study limitations- namely the small numbers particularly in the variant HCL groups- need to be more readily discussed as this is a criticism of the project.
Minor comments:
1. Abstract typo. Should read 7.5% on line 42 page 1
Author Response
Xavier Troussard
Hematologie, CHU Caen Normandie
Caen, France
troussard-x@chu-caen.fr
Caen, April 5, 2022
Dear Editor,
Please find enclosed the new version of the article. The peer reviewing was pertinent and you will find our comments and corrections (in pink), as suggested by the reviewer 3. I deeply thank the reviewer 3 for the comments that improve the quality of the manuscript. We hope the new version is acceptable for publication in Cancers.
Xavier Troussard
Rewiewer 3
The authors have done a nice job of explaining the mutational landscape of hairy cell leukemia and its variants. The have also systemically reviewed all of my comments with adequate responses.
Thank you for your comments that have considerably contributed to improve the quality of the manuscript.
Major discussion
- In the discussion the study limitations- namely the small numbers particularly in the variant HCL groups- need to be more readily discussed as this is a criticism of the project.
We completely agree with the comment. vHCL and SDRPL are rare and the small number of patients is a major limitation of this study. As you advise, we have emphasized this point.
Lines 276-278: We updated and extended the analysis of the mutational landscape of a large cohort of 98 patients with either cHCL or HCL-like disorders, but due of the rarity of the disease only 16 patients account for the HCL-like disorders group (5 patients with vHCL or SDRPL, and 11 patients with HCL-like NOS disease).
Minor comments:
- Abstract typo. Should read 7.5% on line 42 page 1
Thank you,
The mistake was corrected:
Line 42: CDKN1B in 6 pa
